**Technical note: Fu-Liou-Gu and Corti-Peter model performance evaluation for**
**radiative retrievals from cirrus clouds**
S. Lolli[1,2], J. R. Campbell[3], J. Lewis[1], Y. Gu[4], E. J. Welton[5]
[1]NASA GSFC-JCET, Code 612, 20771 Greenbelt, MD, USA
[2] CNR-IMAA, Istituto di Metodologie per l'Analisi Ambientale, Potenza, Italy
[3] Naval Research Laboratory, Monterey, CA, USA
[4] UCLA, Los Angeles, CA, USA
[5] NASA GSFC, Code 612, 20771 Greenbelt, MD, USA
*Corresponding author: slolli@umbc.edu*
**Abstract**
We compare, for the first time, the performance of a simplified atmospheric
radiative transfer algorithm package, the Corti-Peter (CP) model, versus the more
complex Fu-Liou-Gu (FLG) model, for resolving top-of-the-atmosphere radiative
forcing characteristics from single layer cirrus clouds obtained from the NASA Micro
Pulse Lidar Network database in 2010 and 2011 at Singapore and in Greenbelt,
Maryland, USA in 2012. Specifically, CP simplifies calculation of both clear-sky
longwave and shortwave radiation through regression analysis applied to radiative
calculations, which contributes significantly to differences between the two. The
results of the intercomparison show that differences in annual net TOA cloud
radiative forcing can reach 65%. This is particularly true when land surface
temperatures are warmer than 288 K, where the CP regression analysis becomes
less accurate. CP proves useful for first-order estimates of TOA cirrus cloud forcing,
but may not be suitable for quantitative accuracy, including the absolute sign of
cirrus cloud daytime TOA forcing that can readily oscillate around zero globally.

## 1. Introduction

Cirrus clouds play a fundamental role in atmospheric radiation balance and their net radiative effect remains unclear (IPCC 2013; Berry and Mace 2014; Campbell et al. 2016; Lolli et al. 2017). Feedbacks between cirrus dynamic, microphysical and radiative processes are poorly understood, with ramifications across a host of modeling interests and temporal/spatial scales (Liou 1985; Khvorostyanov and Sassen 1998). Simply put, different models parameterize ice formation in varied, yet relatively simplified, ways that impact how cirrus are resolved, and how their macro/microphysical and radiative properties are coupled with other atmospheric processes (e.g., Comstock et al. 2001; Immler et al. 2008). Consequently, models are very sensitive to small changes in cirrus parameterization (Soden and Donner 1994; Min et al. 2010; Dionisi et al., 2013).

Cirrus clouds are the only tropospheric cloud genus that either exerts a positive or negative top-of-the-atmosphere (TOA) cloud radiative forcing effect (CRE) during daytime. All other clouds exert a negative daytime TOA CRE. Cirrus clouds exerting negative net TOA CRE cool the earth-atmosphere system and surface below them. This occurs as the solar albedo term is greater than the infrared absorption and re-emission term. Positive forcing occurs when the two are reversed and infrared warming and re-emission exceed scattering back to space. In contrast, all clouds cause a positive nighttime TOA value, with an infrared term alone and no compensating solar albedo term. This dual property makes cirrus distinct, and why it's crucial to understand how well radiative transfer models are resolving their TOA CRE properties.

The burgeoning satellite and ground-based era of atmospheric monitoring
(Sassen and Campbell 2001; Campbell et al. 2002; Welton et al. 2002; Nazaryan, et
al. 2008; Sassen et al. 2008) has led to a wealth of new data for looking at global
cirrus cloud properties. In particular, TOA CRE, or at the surface (SFC), are evaluated
by means of radiative transfer modeling, designed with different degrees of
complexity. What is not yet known is how the relative simplicity of some models
translates to a relative retrieval uncertainty, given that the CRE effect of cirrus
clouds, at both the ground and TOA, are typically on the order of 1 W m$^{-2}$ (e.g.,
Campbell et al. 2016; Lolli et al. 2017).  Whereas some studies show the relative
uncertainty of such models as static percentages (Corti and Peter, 2009), the
absolute magnitude of uncertainty with respect to cirrus CRE is necessary to
understand whether or not they fit within acceptable tolerance thresholds sufficient
for quantitative use.  Further, given the sensitivity in the sign of net annual cirrus
cloud daytime TOA CRE specifically (Campbell et al. 2016), it's plausible that some
simpler models are routinely aliasing positive versus negative TOA CRE.
Corti and Peter (2009; CP) describe a simplified radiative transfer model that
relies upon a constrained number of input parameters, including surface
temperature, cloud top temperature, surface albedo, layer cloud optical depth, and
the solar zenith angle. CP simplifies drastically the framework of the Fu-Liou-Gu
radiative transfer model (Fu and Liou 1992; Gu et al. 2003; Gu et al., 2011; FLG), for
instance, through a parameterization of the longwave and shortwave fluxes derived
from the FLG model calculations for realistic atmospheric conditions. Moreover, CP
does not directly consider gaseous absorption.  The model has increasingly been
used to assess cirrus cloud radiative effects (Kothe et al. 2011; Kienast-Sjögren et al.
2016; Burgeois et al. 2016) from lidar measurements, owing to its relative simplicity
and lower computational burden compared with a model like FLG.
To date, CP model performance vs. FLG model has been evaluated for
sensitivities only to simulated synthetic clouds and never on real measurements,
especially those collected over long periods (Corti and Peter 2009). Such evaluation,
however, can readily be conducted using the unique NASA Micro Pulse Lidar
Network (MPLNET; Welton et al. 2002; Campbell et al. 2002; Lolli et al. 2013; Lolli
et al., 2014), established in 1999 to continuously monitor cloud and aerosol physical
properties (Wang et al., 2012, Pani et al., 2016).
The objective of this technical note is to then assess differences between CP
and FLG in terms of net annual daytime TOA CRE. CP and FLG model performance
are evaluated using MPLNET datasets collected from Singapore in 2010 and 2011, a
permanent tropical MPLNET observational site, and at Greenbelt, Maryland in 2012,
a midlatitude site. Our goal is to more appropriately characterize the sensitivities of
CP relative to what is generally considered a more complex, and presumably more
accurate, model, with the hopes of better understanding relative uncertainties, and
thus interpreting whether such uncertainties are appropriate for long-term global
cirrus cloud analysis.

**2. Method**
FLG is a combination of the delta four-stream approximation for solar flux
calculations (Liou et al. 1988) and a delta-two–four-stream approximation for IR
flux calculations (Fu et al. 1997), divided into 6 and 12 bands, respectively.  It has
been extensively used to assess net cirrus cloud daytime radiative effects, most
recently for daytime TOA forcing characteristics within MPLNET datasets at both
Greenbelt, Maryland and Singapore, respectively (Campbell et al. 2016; Lolli et al.
2017). The results from these studies have led to the hypothesis of a meridional
gradient in cirrus cloud daytime TOA radiative forcing existing, with daytime cirrus
clouds producing a positive daytime TOA CRE at lower latitudes that reverses to a
net negative daytime TOA CRE approaching the non-snow and ice-covered polar
regions.  They estimate absolute net cirrus daytime TOA forcing term between 0.03
and 0.27 W m$^{-2}$ over land at the mid-latitude site, which ranges annually between
2.20 - 2.59 W m$^{-2}$ at Singapore.  The key here to this phenomenon is the possible
oscillation of the net daytime TOA CRE term about zero, which is believed to vary by
a maximum +/- 2 W m$^{-2}$ in absolute terms (i.e. normalized for relative cirrus cloud
occurrence rate and total daytime percentage locally), after accounting for polar
clouds that should be net cooling elements and varying surface albedos over land
and water exclusively (i.e., not ice).  Resolving such processes thus requires
relatively high accuracy in radiative transfer simulations.

To calculate daytime cirrus cloud radiative effects from MPLNET datasets,

the lidar-retrieved single layer cirrus cloud extinction profile (Campbell et al. 2016;
Lewis et al., 2016, Lolli et al., 2016, Lolli et al., 2017) is transformed into crystal size
diameter (using the atmospheric temperature profile) and ice water content (*IWC*)
profiles using the parameterization proposed by Heymsfield et al. (2014).  Those
parameters, at each range bin, are input into FLG. The thermodynamic atmospheric
profiles, together with ozone concentrations are obtained with a temporal
resolution of +/- 3 hr, from a meteorological reanalysis of the NASA Goddard Earth
Observing System Model Version 5.9.12 (GEOS-5). In contrast, for a given cloud case,
the corresponding cloud and atmospheric CP input parameters are explicitly the
land/ocean surface temperature, the cloud top temperature, the surface albedo, the
cloud optical depth for the specific layer and the solar zenith angle.

Calculations here are performed for the same MPLNET observational sites,

Singapore and Greenbelt, Maryland (i.e., NASA Goddard Space Flight Center; GSFC).
For the former site, two different values of the surface albedo, which is a common
input parameter in both models, are fixed at 0.12 and 0.05, respectively, as
Singapore is a metropolitan area completely surrounded by sea.  This allows us to
more reasonably characterize forcing over the broader archipelago of Southeast
Asia, and follows the experiments described by Lolli et al. (2017). .  At NASA GSFC,
only a single over-land albedo is used, though one that varies monthly between
0.12-0.15 based on climatology.

Here, we reconsider these results by first intercomparing those solved with

FLG and CP for net daytime TOA CRE over a practical range of cloud optical depth
(COD).  As described in both Campbell et al. (2016) and Lolli et al. (2017), daytime is
specifically defined in these experiments as those hours where incoming net solar
energy exceeds that outgoing.  Only under such circumstances can the net TOA CRE
term become negative.  Otherwise, it is effectively nighttime, as the term is positive
and all clouds induce a warming TOA term.  Those nighttime results presented
within the analysis below will instead be considered as context for understanding
net diurnally-averaged differences between the models specifically for the GSFC
dataset.

**3. Intercomparisons**
The daytime cirrus net TOA CRE, normalized by corresponding occurrence
frequency, in this case as a function of COD, was evaluated at Singapore (1.3 N, 103.8
E, 20 m above mean sea level) and GSFC (38.9 N, 76.8 W, 39 m above mean sea
level) for both FLG and CP.  The method to estimate MPLNET cirrus cloud optical
properties is described in Lewis et al. (2015) and Campbell et al. (2016), for both 20
and 30 sr solutions from the unconstrained single-wavelength elastic lidar equation
at 532 nm (Campbell et al. 2016).   The latter constraint provides "bookend"
estimates for TOA CRE designed to approximate system variance.  For both models,
the daytime cirrus cloud net TOA CRE is calculated as the difference of two model
computations using different assumed states (cloudy sky minus cloud and aerosol
particulate-free conditions) to isolate the distinct cirrus cloud impact alone (in W m$^{-}$
$^{2}$).
*3.1 Model sensitivities*
An initial sensitivity study was carried out to evaluate how the input
parameters, and eventually their uncertainties, influence the net TOA CRE
calculations. Results are summarized in Table 1.   Model input parameter
sensitivities were investigated for surface albedo, COD, land/ocean surface
temperature and cloud top temperature.  Table 1 shows how much net, SW and LW
fluxes change by varying each individual parameter alone.  For instance, changing
the surface albedo from 0.12 to 0.14 and keeping the other three parameters fixed
produces similar changes in both models (26% for CP model and 25% for FLG
model). Changing COD from 1 to 1.1 produces a change of 16% for CP and 21% for
FLG. Changing surface temperature and cloud top temperature of 1K produces
respective changes of 10% and 7% for CP and 7% and 6% for FLG. Though subtle,
the models exhibit some differences in variance relative to the input parameters
required to initialize them.

*3.2 Singapore (2010-2011)*

FLG and CP were compared over a total of 33072 total daytime single layer

cirrus clouds at Singapore from 2010 to 2011. Figures 1, 2, 3 and 4 reflect
histograms of cirrus cloud relative frequency and net annual daytime TOA CRE
normalized by corresponding frequency, for both surface albedo values of 0.05 (Fig.
3 and 4; i.e., over sea) and 0.12 (Fig. 1 and 2; i.e. over land) at 0.03 COD resolution
from 0 to 3. This latter COD range was chosen to distinguish cirrus clouds in a
phenomenological manner consistent with Sassen and Cho (1992). Note, since a
common cloud sample is used, the 20 sr samples vary in COD between only 0 and
approximately 1 in contrast to the 30 sr sample topping out at 3. The observed
differences in net radiative effect can be ascribed to the different lidar ratio. Overall,
the results here complement the work of Berry and Mace (2014), who first
recognized the significance of optically-thin cirrus influencing the net normalized
term so significantly.
Intercomparison of net daytime TOA CRE vs. COD over the ocean at 30 sr, we
obtain -0.89 W m$^{-2}$ from CP and -0.37 W m$^{-2}$ for FLG. The overall CP net TOA CRE is
greater in absolute magnitude than FLG by a maximum difference of 58%. This
value is obtained by taking the ratio between yearly CRE from FLG over CP and then
the percentage difference. Over land (urban environment), CP net daytime TOA CRE
are higher than the FLG model by 25% (CP=4.43 W m$^{-2}$ and FLG=3.35 W m$^{-2}$). The
COD value at which cirrus begin cooling the earth-atmosphere system, moving
toward higher COD, is systematically shifted towards higher values for CP with
respect to FLG.
To better understand the different outputs between the two models, a scatter
plot between from FLG barplot entries is shown in Figs. 2 and 4, and the
corresponding CP barplot values are plotted, over land and over ocean, in Figs. 5
and 6. The blue line represents the actual linear data regression, while the red line
represents an ideal case (i.e., slope=1, intercept=0).  If the two radiative transfer
models show identical results regarding CRE, all the points should lie on the blue
line. The red line instead represents the actual regression line, or a relative measure
of how much the two models differ.
From Figs. 5 and 6, the FLG-derived net daytime CP TOA CRE values are
systematically greater in absolute value than the corresponding FLG values by 60%.
More in detail CP TOA CRE of 1 Wm$^{-2}$ corresponds with FLG values ranging from
0.57 Wm$^{-2}$ to 0.59 Wm$^{-2}$. On the contrary, the bias (or the intercept from the linear
regression) shows higher variability depending on the surface type underlying the
cirrus cloud (land versus ocean). This indicates that when a cirrus cloud shows a
neutral effect (0 Wm$^{-2}$) for CP model,  FLG model solutions range from -0.05 (land)
to -1.1 Wm$^{-2}$ (ocean),. This implies that characterization of cirrus cloud warming or
cooling effects depend on the model.

For the sake of completeness, and to cover all the variability related to the

chosen LR, we performed the same analysis though excluding the 20 sr solution.
Over the ocean, we derive an overall forcing of 1.34 W m$^{-2}$ for CP and 0.48 W m$^{-2}$ for
FLG (41%). In Fig. 3 (blue arrow), a shift is clearly evident near 0.25 COD (0.6 for CP
and 0.35 for FLG) in CRE sign change (from positive to negative).  Over land, we
estimated CP = 4.20 W m$^{-2}$ and FLG=2.98 W m$^{-2}$.(68%)

*3.4 Greenbelt, Maryland 2012*

To limit potential assessment ambiguity based on a single-site analysis, we

performed a second model comparison using the 2012 NASA GSFC dataset. A
summary of this dataset and net daytime TOA CRE results can be found in Campbell
et al. (2016). As this site in land-locked, only the single albedo was, again, used,
though varied monthly based on climatological passive satellite estimates. 21107
daytime cirrus cloud profiles were considered.  Shown in Figure 6 (upper panel) are
the total net TOA CRE vs. COD at 30 sr, for CP (-2.59 Wm$^{-2}$) against FLG (0.05 Wm$^{-2}$).
A relative differencing here is impractical.  Suffice however, this is a significant
difference, and the sign of the net daytime forcing term is uncertain when
comparing the two.

With this NASA GSFC dataset, we further consider an additional 32185

nighttime cirrus cloud cases within the analysis (Fig. 6, lower panel).  Relative to
prior estimates of CP uncertainty compared with more complex models, a diurnal
average would be likely to produce a different, and plausibly closer, relative
agreement consistent with prior studies.  That is, since during for most of the period
we define here as night there is no solar input, a simplification of the infrared
forcing terms and parameterizations alone would potentially yield a closer
comparison between the two models.  For the NASA GSFC dataset, we solved a
relative net nighttime TOA CRE of 29.1 $Wm^{-2}$ with FLG compared with 21.0 $Wm^{-2}$
with CP, or a relative difference approaching 50%.. Summarized in Table 2 are the
discrepancies in terms of CRE at both observational sites.

It is useful at this point to discuss some of the potential elements driving

these differences.  The larger discrepancies between the two models are likeliest
ascribed to the parameterization of three specific parameters in the CP model: the
first two, $\sigma^*$ and $k^*$ (Eq. 2 of Corti and Peter, 2009) are two approximated
parameters for the Stefan-Boltzmann constant and the surface temperature
exponent estimated from radiative calculations and used to calculate the outgoing
longwave earth radiation. The last parameter, $\gamma^*$ (again obtained from a regression
analysis), is related to the asymmetry factor of cloud droplets and used to calculate
the cloud reflectance of shortwave radiation (Eq. 11 in Corti and Peter; 2009).  We
speculate that, though the analysis is left to a future study on broader uncertainties
in modeling ice radiative properties inherently with any model, these parameters
are not the constants ascribed by CP, but that their values instead change with
respect to season and latitude.
The 20% relative model accuracy claimed in Corti and Peter (2009) may be
verified for special conditions in tropical latitudes, where the three parameters
discussed above are well optimized.  But, that is clearly not found from our study.
Corti and Peters (2009) expressly stated that they used fixed values for those three
parameters (i.e., $\sigma^*$ and $k^*$ in Eq. 2 and $\gamma^*$ in Eq. 11 in Corti e Peter, 2009) again
using regression analysis, but this shouldn't be the case, as net TOA CRF is very
sensitive to those parameters.  For example, varying water vapor concentrations in
the atmosphere can be the responsible of a difference up to 25 $Wm^{-2}$ (for
temperatures at the surface higher than 288K) in clear-sky earth longwave radiation
at Singapore, as stated in Corti and Peter (2009; Fig. 1). In our analysis we verified
that, over one year, the land surface temperature is higher than 288K 66% of the
time.  For this reason, to assess if land surface temperature is responsible for these
larger discrepancies, we reproduced Fig. 6 (upper panel) masking out all cases
corresponding with land surface temperatures higher than 288 K at Greenbelt (in
Singapore those temperatures are mostly during nighttime). Shown in Figure 7 are
the results of the analysis. CP and FLG radiative transfer models in this range of
temperature are in much better agreement (NET CP = -8.06 $Wm^{-2}$; NET FLG -8.65
$Wm^{-2}$), within 6%. Choosing surface temperatures lower than 288 K is reducing the
temperature gradient between the surface and cloud top, limiting the cloud thermal
warming effect (see Eq. 6 of Corti and Peter, 2009). Moreover, lower temperatures
are usually associated with higher solar zenith angle that implies stronger albedo
effects. For this reason, in Fig. 7 the albedo effect is outweighing the capacity of
cirrus cloud in trapping longwave radiation, with a net cooling effect estimated.

We advise that those looking to apply CP to long-term climate/cirrus cloud

study should carefully analyze the relevance of these settings to their given
experiment before directly applying the model, especially when land surface
temperatures are warmer than 288K.
**4. Conclusions**

Annual single-layer cirrus cloud top-of-the-atmosphere (TOA) radiative

effects (CRE) calculated from the Corti and Peter (2009) radiative transfer model
(CP) are compared with similar results from the more complex, and presumably
more accurate, Fu-Liou-Gu (FLG) radiative transfer model. The CP model calculates
CRE using a parameterization of longwave and shortwave fluxes that are derived
from real measurements optimized for a tropical environment through a regression
analysis to simplify the radiative calculations.  Values for these parameterizations,
as suggested in Corti and Peter (2009), lead to relative differences in TOA CRE that
far exceed the stated 20% in the original manuscript. This includes parsing results
out for daytime, nighttime or diurnal averages.  It is believed that specific
parameterizations with the simplified model cannot be considered global constants,
as originally defined for CP, but that they should be carefully evaluated on single
case basis for each experiment. Moreover we find that the land surface temperature
is responsible for significant discrepancies when larger than 288K, because the
original CP regression analysis is less accurate for larger temperatures. However, CP
uses less input parameters compared with FLG, making it practically and
computationally more efficient, particularly for large climate datasets.  This is the
first time, however, that the two models are compared using long-term cirrus clouds
datasets, as opposed to synthetic datasets, with experiments conducted using NASA
Micro Pulse Lidar datasets collected at Singapore in 2010 and 2011 (Lolli et al.
2017) and Greenbelt, Maryland in 2012.

Net daytime TOA CRE was evaluated versus cloud optical depth (COD) for

steps of 0.03 (COD range: 0-1) at 20 sr and for steps of 0.1 at 30 sr (COD range: 0-3)
for Singapore datasets, while at 30 sr for Greenbelt, Maryland. Our findings suggest
that the difference in annual net TOA CRE between the two models approaches 65%
in one experiment at Singapore.  At Greenbelt, Maryland, the sign of the net annual
daytime TOA CRE term differs, and the absolute difference varies between by nearly
2.5 Wm$^{-2}$.  Differences in the sign of the net TOA forcing term, however, are most
worrying.  Since cirrus clouds are the only cloud that can exhibit daytime positive or
negative net TOA CRE, subtle differences in absolute magnitude are less important
than whether or not the clouds are inducing a cooling or forcing term in the TOA
radiation budget.

In spite of this comparison, even if we reasonably speculate that FLG is the

more accurate model overall, because of its relative complexity compared with CP,
we are still missing regular comparisons of FLG with real observational data.  Thus,
the practical gains to long-term application of a simplified model like CP cannot be
overstated, given lower computational demands.  However, we believe that the
results from this study are noteworthy because they show that the differences
between the two models are significant. With respect to cirrus annual net daytime
TOA CRE, and given the perspective on their global distribution described by
Campbell et al. (2016) and Lolli et al. (2017), these sensitivities can lead to
completely different conclusions about global cirrus TOA forcing effects.  Therefore,
in future work, it is imperative on the community to continue understanding and
refining the global parameterizations used in all radiative transfer models regarding
cirrus.  Continued intercomparisons between models with real observation both at
ground (using flux measurements), in situ (aircraft measurements) and at TOA
(using satellite-based measurements,) remain critical interests.

**Acknowledgements**
This study and the NASA Micro Pulse Lidar Network (MPLNET) are supported by
the NASA Radiation Sciences Program (H. Maring).  Author JRC acknowledges the
Naval Research Laboratory Base Program (BE033-03-45-T008-17) and support of
NASA Interagency Agreement NNG15JA17P on behalf of MPLNET.

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

107, 8019,


**FIGURES**

**FIGURE 1** Analysis over land (Albedo=0.12) for 20sr solution. CRE vs. COD is
weighted by occurrence frequency for Corti and Peter(red) and FLG
(blue) models over 2010-2011

**FIGURE 2** Analysis over land (Albedo=0.12) for 30sr solution. CRE vs. COD is
weighted by occurrence frequency for Corti and Peter(red) and FLG
(blue) models on 2010-2011.

**FIGURE 3** Same as Figure 1, but over the ocean (Albedo=0.05). The arrow shows the
shift in COD for CRE sign change between the two models

**FIGURE 4** Same as Figure 2, but over the ocean (Albedo=0.05)

**FIGURE 5** Scatter plot and linear regression for 30sr solution for FLG and CP CRE in
2010-2011 over land (upper panel) and ocean (lower panel)

**FIGURE 6** Analysis on 2010 dataset from MPLNET GSFC observational site for 30sr
solution daytime (upper panel) and nighttime (lower panel).

**FIGURE 7** Same as Figure 6, taking out those measurements with a land surface
temperature $T_{surf}$ > 288K

Tables

| NET CP | NET FLG | LW TOA FLG | SW TOA FLG | LW TOA CP | SW TOA CP | |
|--------|---------|------------|------------|-----------|-----------|------|
| -12.6 | -9.4 | 67.8 | -77.2 | 69 | -81.6 | Ref |
| 9.3 (26%) | -7 (25%) | 67.8 | -74.8 | 69 | -78.3 | Albedo |
| -14.7 (16%) | -11.4 (21%) | 71.8 | -83.2 | 73.5 | -88.2 | Cod |
| -11.3(10%) | -8.7(7%) | 68.5 | -77.2 | 70.3 | -81.6 | Surf Temp |
| -13.5(6%) | -10(5%) | 67.2 | -77.2 | 68.1 | -81.6 | Cl Top Temp |

Table 1 Total NET, SW and LW fluxes (W/m²) at TOA. Sensitivities of CP and FLG
radiative transfer models with respect to the surface albedo, cloud optical depth
Unperturbed parameters are COD=1, Surface albedo=0.12, $T_{surf}$=294K Cloud top
$T_{top}$=229K. The variation in net radiative forcing expressed in percentage for each
parameter are calculated changing the surface albedo from 0.12 to 0.14, the COD
from 1 to 1.1, and augmenting the temperatures of 1K.

| CRE vs. COD | Ocean | Land |
|-------------|-------|------|
| SING 2010-2011 | 20sr CP=1.34 FLG=0.48(65%) 30sr CP=-0.89 FLG=-0.37 (58%) | 20sr CP=4.20 FLG=2.98 (30%) 30sr CP=4.43 FLG=3.35 (25%) |
| GSFC 2012 | | 30sr CP=-2.59FLG=0.07 |

Table 2 Summary of principal CRE (Wm⁻²) differences between FLG and CP radiative
transfer model depending on year and on land/ocean.

Figures

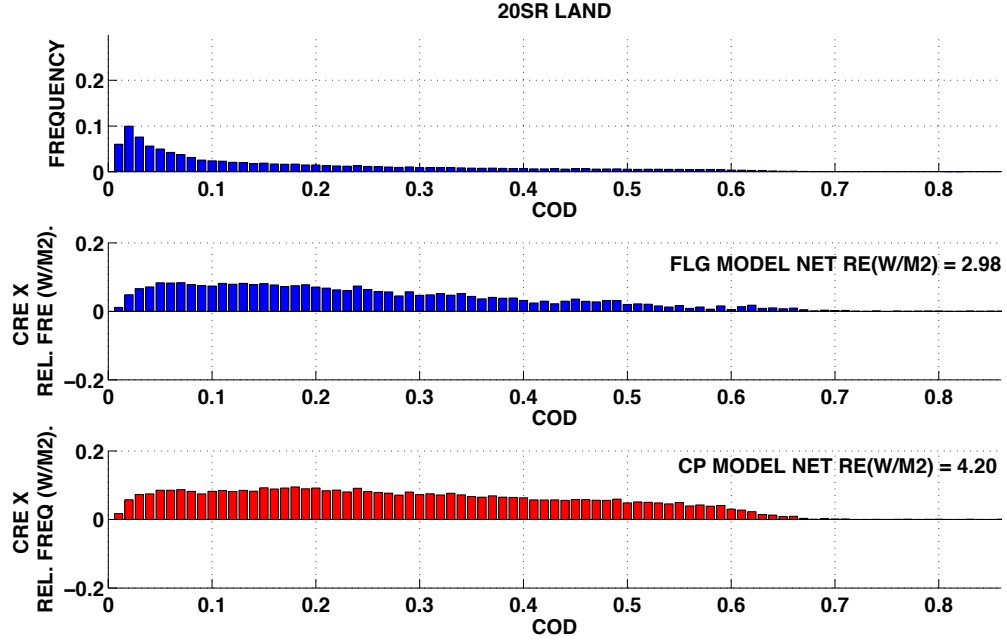


**Figure 1** Analysis over land (Albedo=0.12) for 20sr solution. CRE vs. COD is weighted by occurrence
frequency for Corti and Peter(red) and FLG (blue) models over 2010-2011

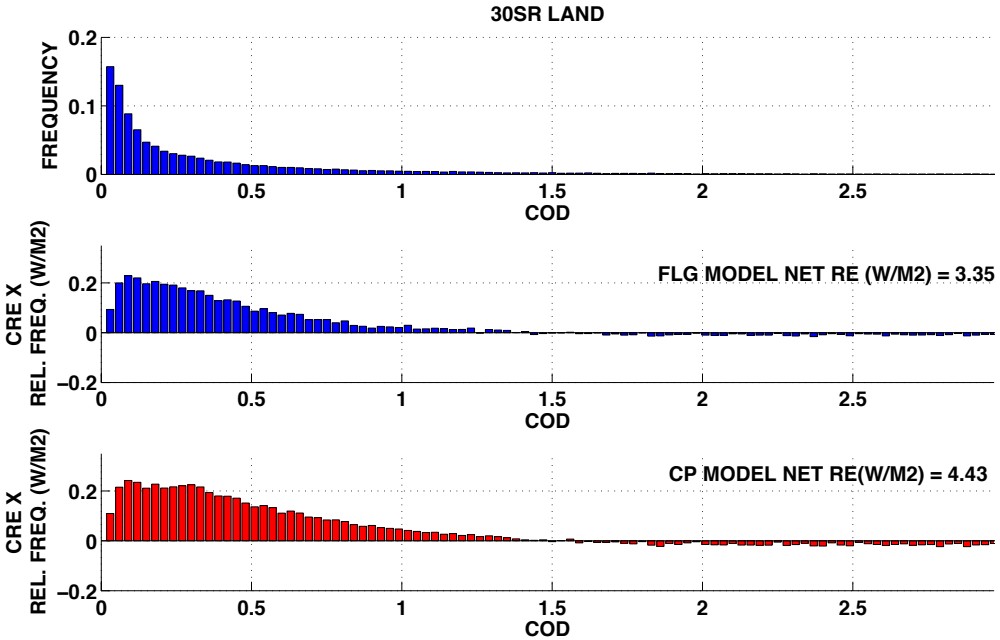


**Figure 2** Analysis over land (Albedo=0.12) for 30sr solution. CRE vs. COD is weighted by occurrence
frequency for Corti and Peter(red) and FLG (blue) models on 2010-2011.

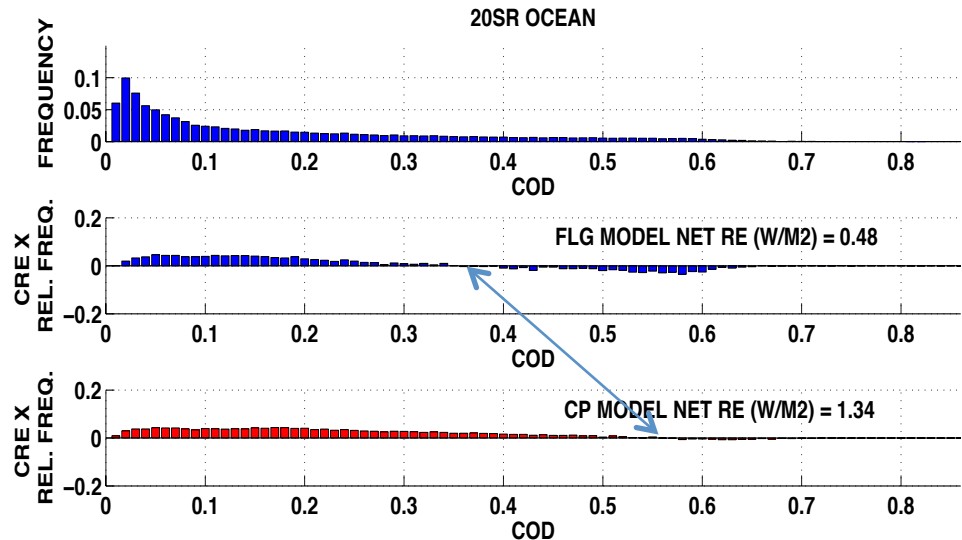

**Figure 3** Same as Figure 1, but over the ocean (Albedo=0.05) . The arrow shows the shift in COD for
CRE sign change between the two models

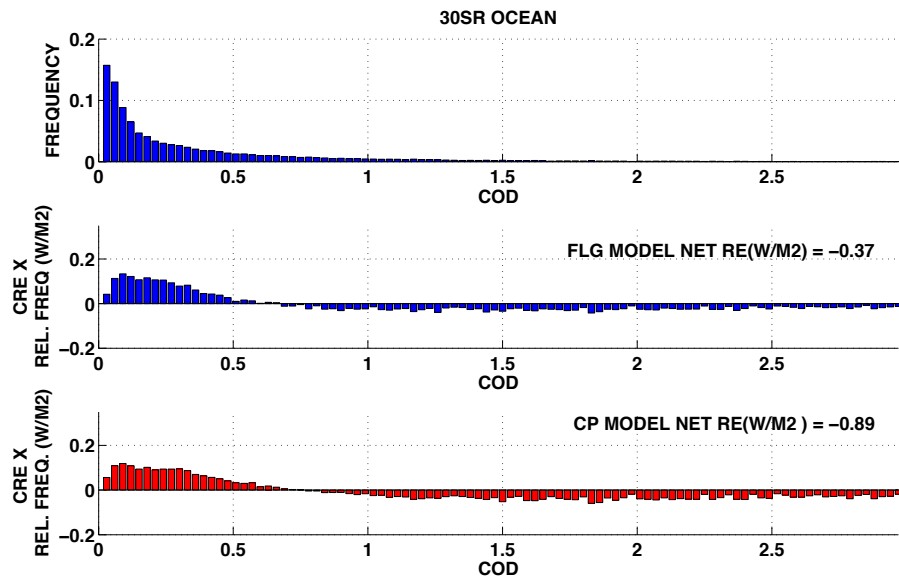


**Figure 4** Same as Figure 2, but over the ocean (Albedo=0.05)


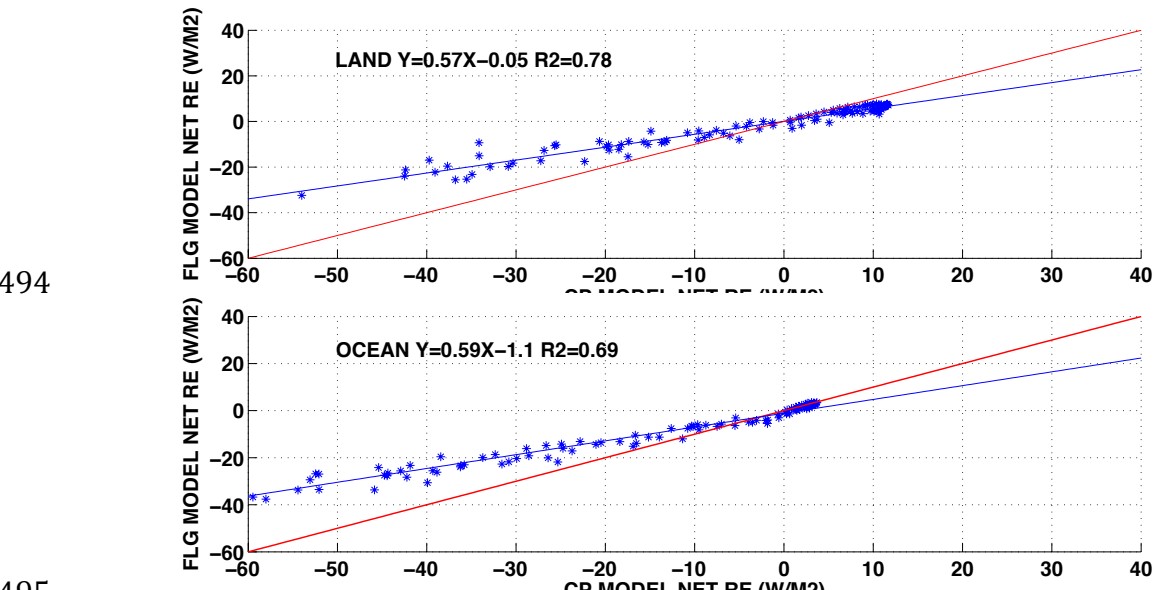

**Figure 5** Scatter plot and linear regression for 30sr solution for FLG and CP CRE in 2010-2011 over
land (upper panel) and ocean (lower panel)

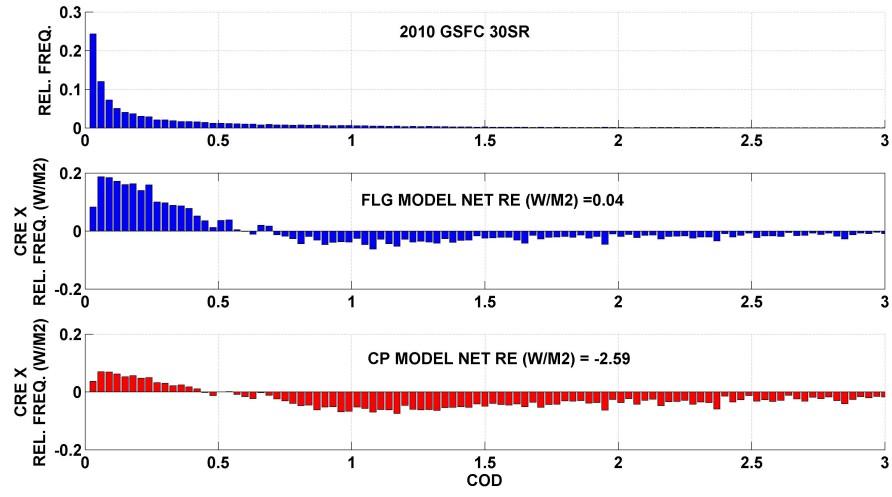


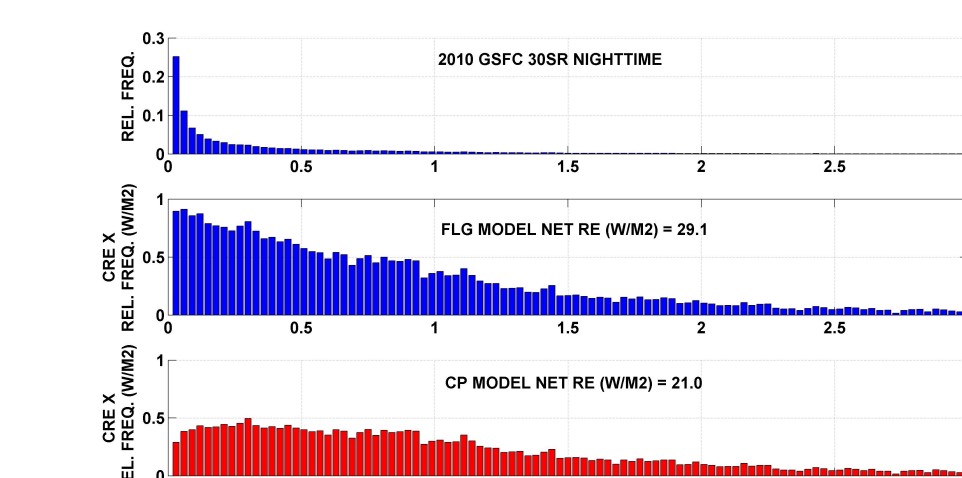

**Figure 6** Analysis on 2010 dataset from MPLNET GSFC observational site for 30sr solution daytime
(upper panel) and nighttime (lower panel).

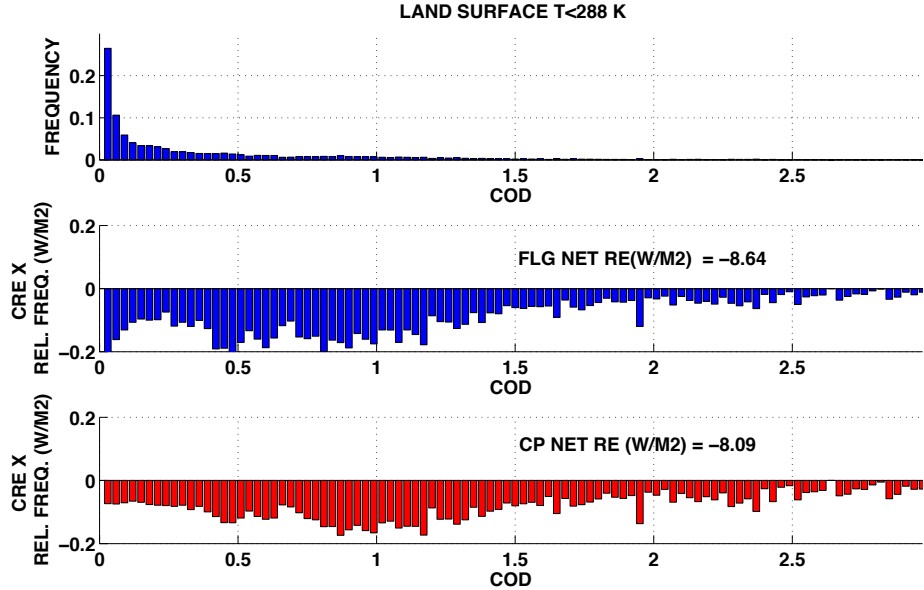

Figure 7 Same as Figure 6, taking out those measurements with a land surface temperature $T_{surf} >$
288K