# Peer review of "Technical note: Fu-Liou-Gu and Corti-Peter model performance evaluation for"

_Atmospheric Chemistry and Physics, 2016_

## Referee Comment (RC1) · Anonymous Referee #1 · 29 Dec 2016

1. Error bars should be added to the plots, such as how large the uncertainty is due to the uncertainty in the input variables? 2. How does the magnitude of the disagreements between two models compare with these error bars? 3. Missing of validation: validation for this intercomparison study is necessary. ARM data can be used for this purpose where both cloud and radiation measurements are available.

---

## Referee Comment (RC2) · Anonymous Referee #2 · 6 Jan 2017

**Review of Lolli et al. manuscript**

This is a relevant short study pointing out (to my knowledge the first time) the single-layer cirrus cloud radiative effect differences between the Fu-Liou-Gu radiative transfer model (FLG) calculations and the simplified Corti and Peter (CP) model, published in the Corti and Peter 2009 paper (CP2009). Due to its simplicity and supposedly quite accurate results, the CP model has in recent years been used in several studies (e.g. Bourgeois et al., 2016, Kienast-Sjörgren et al., 2016).

Lolli et al.'s manuscript take data from 2 years of lidar dataset from Singapur and calculate considerably larger differences between the more advanced FLG and CP radiative models compared to what reported in CP2009.

In general, I am missing a more detailed discussion on the bias sources by the CP model and a more careful comparison with CP2009 (day only vs daily mean conditions).

The manuscript would increase its scientific significance if a midlatitude dataset, for instance the one published in Campbell et al., 2016, would be added to the analysis (or at least discussed the possible implications for regions outside the tropics, where CP model has also been used).

In general, I find this technical note valuable, however, the authors need to address the listed comments/questions before the paper is published in ACP.

**General comments/questions:**

1.)
Is the Singapore lidar site representative for the tropics?
(being an urban site, in a polluted region, etc.)
Would the main conclusion change, for instance, when applying the radiative transfer model calculations to other tropical measurement locations of the MPLN network e.g. the Bermudas, Cabo Verde, Doi Ang Khang (Thailand), Douliu (Taiwan), EPA-NCU (Taiwan), Kanpur (India), Ragged Point (Barbados), etc.?

2.)
Please add/estimate the uncertainty of both your absolute cloud radiative effect calculations as well as its deviations from the CP model.

3a.)
I miss a more detailed discussion of differences between CP and FLG models. Lines 183-185 and 207-209 need to be expanded.
Can you somehow test this speculation? Could you remove the longwave absorption above clouds from FLG model (or add this calculation to the CP model) and confirm the hypothesis?

Please add or at least comment on the uncertainty estimates of the radiative calculations
(see also a related specific comment 7).

3b.)
Did you explain the reasons for significantly different intercepts in figures 5 and 6?
lines 175-185: I don't understand the interpretation of the different intercept parameter. Please rephrase.
Are the lines 183-185 referring in general (that is – not only intercept parameter) to differences

between CP and FLG models or do they refer to the intercept parameter only?

4.)
CP2009 uses the daily mean conditions to asses their radiative transfer model. Would taking into account both day and night data decrease the bias/bring your results closer to the bias of up to 20% as stated in CP2009?
Consequently, can you comment/calculate how would differences between CP and FLG radiative model calculations behave during night?

5.)
What is the additional information we gain by always having 2 years of data shown in Figures 1-4 in separate panels?
One figure where both years are shown separately can be to my quick judgment followed by a combined histograms for both years.
Please also comment whether 2 years of data are enough to get a reasonable "climatological" values?
Why the authors didn't use the whole available Singapore lidar dataset?

**Specific comments:**

1. Please state your definition of cirrus clouds in a condensed form for the convenience of the reader
2. line 50: 1 W m$^{-2}$
3. line 67: Bourgeois et al., 2016 does not appear in reference list
4. line 110: How much can GEOS-5 biases influence the results?
5. line 157: "*This is particularly evident over ocean at 20sr...*" Why?
6. line 203: *in *
7. line 214: "*...given lower computational demands…*" Can you quantify that?
8. lines 218-219: this is a strong statement for a study that analyzes only 1 site!

9. Please expand figures 1-4 in y-direction, so that one can better read out the values
10. Use a reasonable number of digits after the dot for the NET RE value in figures 1-4, and use them consistently with those stated in text.
11. Figure 1: top and bottom panel do not have the same upper y-axis limit
12. Could you briefly comment on the net negative daytime TOA CRF cooling effect for the thinnest cirrus clouds as observed in Campbell et al. 2016? Why you do not/cannot see that from your dataset?

**References:**

Corti and Peter, 2009: A simple model for cloud radiative forcing
Bourgeois et al., 2016: Ubiquity and impact of thin mid-level clouds in the tropics
Kienast-Sjörgren et al., 2016: Climatological and radiative properties of midlatitude cirrus clouds derived by automatic evaluation of lidar measurements
Campbell et al., 2016: Daytime Cirrus Cloud Top-of-the-Atmosphere Radiative Forcing Properties at a Midlatitude Site and Their Global Consequences

---

## Author Comment (AC1) · 4 Mar 2017

ACP-2016-980

The authors gratefully acknowledge the suggestions of the two peer reviewers assigned to this manuscript. We further thank the Associate Editor for dutifully working to have competent and constructive reviews for the paper. We thank them for their service to the journal.

Thanks again to all.

Reviewer Comments indented in red
Author Responses in black

Reviewer #1

1. Error bars should be added to the plots, such as how large the uncertainty is due to the uncertainty in the input variables?

The main objective of this technical note is to compare single-layer cirrus cloud radiative forcing calculated by the Fu-Liou-Gu radiative transfer model and by the Corti-Peter model. Thus, our objective is pursued by inputting into the two models the same cloud optical properties, the same thermodynamics of the atmosphere and surface albedo. As the input parameters are the same for both models, it follows that the uncertainty associated with these input variables is the same. Differently, the Fu-Liou-Gu model needs much more parameters. Some of them are obtained from atmospheric models, like the ozone or $CO_2$ concentrations and it is objectively difficult to assign an uncertainty to those variables.

Nevertheless we followed the reviewer's suggestion and added in the manuscript a sensitivity study (Par 3.1) for the common variables, or how the net radiative forcing calculated by the two models changes in percentage with respect to the common input variables (Table 1)

2. How does the magnitude of the disagreements between two models compare with these error bars?

See previous answer.

3. Missing of validation: validation for this intercomparison study is necessary. ARM data can be used for this purpose where both cloud and radiation measurements are available.

Because of the Corti-Peter model, the intercomparison is done at the Top of the Atmosphere, where no ARM data are available. Its presently unclear whether or not CERES-type analysis could be performed in a manner suggested by the reviewer, given the highly diffuse nature of many cloud samples analyzed. That remains a topic for another study and analysis.

Thank you for your careful reading of the manuscript.

Reviewer #2

Reviewer #2

This is a relevant short study pointing out (to my knowledge the first time) the single-layer cirrus cloud radiative effect differences between the Fu-Liou-Gu radiative transfer model (FLG) calculations and the simplified Corti and Peter (CP) model, published in the Corti and Peter 2009 paper (CP2009).

Thank you very much for the positive comment, and for the very thorough reading of the paper. Many positive changes have been made as a result of your comments.

Due to its simplicity and supposedly quite accurate results, the CP model has in recent years been used in several studies (e.g. Bourgeois et al., 2016, Kienast-Sjörgren et al., 2016).

Lolli et al.'s manuscript take data from 2 years of lidar dataset from Singapore and calculate considerably larger differences between the more advanced FLG and CP radiative models compared to what reported in CP2009.

In general, I am missing a more detailed discussion on the bias sources by the CP model and a more careful comparison with CP2009 (day only vs daily mean conditions).

We appreciate the thought. We've added qualitative interpretation of where such differences arise. We found out that the main problem related to three parameters obtained through a regression analysis and set up as a constant. This is not the case, and probably those parameters should be optimized regarding the specific analysis the model is performing. Those parameters influence both longwave and shortwave calculations. In CP radiative model those values are set as constant. No information more is available. We found out that the results are very sensitive to those variables. Text has been added in Abstract , Par 3.4 and Conclusions.

The manuscript would increase its scientific significance if a midlatitude dataset, for instance the one published in Campbell et al., 2016, would be added to the analysis (or at least discussed the possible implications for regions outside the tropics, where CP model has also been used).

We agree with the reviewer and have added the analysis from GSFC in the manuscript (Par. 3.4)

In general, I find this technical note valuable, however, the authors need to address the listed comments/questions before the paper is published in ACP.

General comments/questions:
1.)
Is the Singapore lidar site representative for the tropics? (being an urban site, in a polluted region, etc.) Would the main conclusion change, for instance, when applying the radiative transfer model calculations to other tropical measurement locations of the MPLN network e.g. the Bermudas, Cabo Verde, Doi Ang Khang (Thailand), Douliu (Taiwan), EPA-NCU (Taiwan), Kanpur (India), Ragged Point (Barbados), etc.?

Its unclear exactly how relevant the comment is. As we've now added GSFC, there is a second site from which to consider the model differences. But, of course, they are relative differences. Thus, it wouldn't matter what site(s) we ultimately picked. But, it was a good suggestion adding GSFC to perhaps suppress any ambiguity in our conclusions based on the single-site analysis.

2.)
Please add/estimate the uncertainty of both your absolute cloud radiative effect calculations as well as its deviations from the CP model.

Thanks. A sensitivity study to the input parameters for both models has been done and introduced in the new manuscript (Par. 3.1 and Table 1)

3a.)
I miss a more detailed discussion of differences between CP and FLG models. Lines 183-185 and 207-209 need to be expanded.

Can you somehow test this speculation? Could you remove the longwave absorption above clouds from FLG model (or add this calculation to the CP model) and confirm the hypothesis? Please add or at least comment on the uncertainty estimates of the radiative calculations (see also a related specific comment 7).

We changed completely the manuscript and we introduced the new findings on why those discrepancies arise, in abstract Par. 3.4 and conclusions.

3b.)
Did you explain the reasons for significantly different intercepts in figures 5 and 6?

The intercepts in Fig. 5 and 6 are different because there is a bias introduced by the three parameters obtained through a regression analysis. This is now stated in the text.

lines 175-185: I don't understand the interpretation of the different intercept parameter. Please rephrase. Are the lines 183-185 referring in general (that

is – not only intercept parameter) to differences between CP and FLG models or do they refer to the intercept parameter only?

The text has been changed accordingly. Now lines 183-185 are part of another paragraph making the manuscript clearer.

4.)
CP2009 uses the daily mean conditions to asses their radiative transfer model. Would taking into account both day and night data decrease the bias/bring your results closer to the bias of up to 20% as stated in CP2009? Consequently, can you comment/calculate how would differences between CP and FLG radiative model calculations behave during night?

Taking into account only nighttime means to compare only the LW outgoing radiation. We believe that the reviewer is right, and we add this intercomparison to the manuscript to check if the bias is more evident in some bands with respect to the others (Figure 7)

5.)
What is the additional information we gain by always having 2 years of data shown in Figures 1-4 in separate panels? One figure where both years are shown separately can be to my quick judgment followed by a combined histograms for both years. whether 2 years of data are enough to get a reasonable "climatological" values? Why the authors didn't use the whole available Singapore lidar dataset?

We agree that grouping the pictures is saving space giving the same amount of information. Nevertheless, due to the barplot properties under MATLAB, grouping the pictures is generating confusion.

Those two years investigated were the best/most complete years in the archive, due to instrument failures/swapouts.  We appreciate the thought.

**Specific comments:**

1. Please state your definition of cirrus clouds in a condensed form for the convenience of the reader

Done

2. line 50: 1 W m-2

Done

2. line 67: Bourgeois et al., 2016 does not appear in reference list

Added

3. line 110: How much can GEOS-5 biases influence the results?

Actually there is no influence in using GEOS-5 model or actual radiosounding data, as the same temperatures are inputted in the two models

4. line 157: "This is particularly evident over ocean at 20sr…" Why?

This is a direct consequence of Figure 6, as we can notice a larger bias between the two models (the discrpance between blue and red lines). As a consequence results are shifted with respect to the COD (CRE with a COD = 0.4 for FLG has the same effect of COD = 0.2 for CP)

6. line 203: in in

Fixed

7. line 214: "…given lower computational demands…" Can you quantify that?

CP model analyzes one year dataset in less than 5 minutes while FLG needs    > 24 hours. The info has been added in the manuscript

8. lines 218-219: this is a strong statement for a study that analyzes only 1 site!

We agree that the statement is strong, but now there is evidence from a second site too

9. Please expand figures 1-4 in y-direction, so that one can better read out the values

We expanded the scale, but the format of the picture is standard

10. Use a reasonable number of digits after the dot for the NET RE value in figures 1-4, and use them consistently with those stated in text.

Fixed

11. Figure 1: top and bottom panel do not have the same upper y-axis limit

Fixed

12. Could you briefly comment on the net negative daytime TOA CRF cooling effect for the thinnest cirrus clouds as observed in Campbell et al. 2016? Why you do not/cannot see that from your dataset?

This is a direct consequence of the existence of a meridional gradient in cloud radiative effect. Being close to the equator, even the thin cirrus cloud are keen to warm the system earth-atmosphere.

Thank you again for your comments and careful scrutiny. We appreciate your consideration.

---

## Referee Report (RR1)

**Review of Lolli et al., ACP -2016-980**

I applaud at the inclusion of the new midlatitude dataset to the comparison, as well as a better evaluation of the origin of discrepancies between the FLG and CP models.  However, I still think the authors stop just a step short of going to the bottom of the problem. Also, separating the CRE into the LW and SW components would help in understanding the results better in some of the manuscript sections.

**Comments**

1.)Lines 17-18: "Specifically, CP simplifies…LW…"

You don't mention the simplification of the SW CRE calculation here, despite referring to the parameter $y*$ (related to simplifications of SW flux calculations in CP model) further in the text?

2.) You might want to mention the Greenbelt site in the abstract.

3.) Chapter 3.1 – model sensitivities for -> daytime only? Please specify!
It would be also valuable to include information on LW and SW CRE component, not only NET.
I think that should be done already in this study (considering its simplicity), and not simply left as an outlook (which you state in lines 310-313).

4.) Chapters 3.2 and 3.3
I don't see what is the advantage of having both years separately, and not clustering the results together. If you keep them separately, you need to understand what caused a different value of the CP model bias (suggestion: Does it correlate with the surface temperature?)

And – if you keep two separate panels  - Were the 2 years, meteorologically significantly different? Different ENSO phase, etc.?

5.) Lines 205 – 214 :
I would suggest first stating the general conclusion (CP larger values for 40-60%) , followed by an example (1 W m$^{-2}$ vs. 1.4-1.6 W m$^{-2}$) .

"..shows higher variability depending on the year…"
Yes, but year isn't the root cause of the change. What was different between 2010 and 2011? Maybe surface temperature?

Lines 213-214: What do you mean by "must carefully be determined with these models"
Which models? CP? FLG? Other?

6.) Lines 235-236:

I don't see any significant differences in the error between the 2 sites.

7.) Mention why CP2009 used $\sigma^*$ and $k^*$ and not simply $k$ and $\sigma$.

8.) Lines 247-250:
Making a step further and trying to understand better what exactly is causing the error is something one could quickly look at. For instance, based on the Fig. 1 in CP2009 and your text lines 261-265, one would expect the error to be related to temperature, at least in the tropics. Your current conclusions of Chapter 3.4 do not add much to what already stated in CP2009.

Therefore I would suggest (at least) one new sensitivity test in which you would mask out lidar measurements in Singapore with T>288 K and try to understand if that leads to a better agreement with FLG model.  (or select measurements at warmest/coldest surface temperature conditions for both Singapore and Greenfield).

This sensitivity test would be useful in better determining the conditions at which the CP model's bias is still at acceptably low levels (maybe those stated in CP2009 paper?).

9.)
Figures 1-6:
I still don't see why not making a technical effort with Matlab to cluster the years 2010 and 2011. Couldn't you just make a sum of the two histograms/bar plots, as the x-axis values do not change?
This would sharpen the main points of the paper.

**Minor comments**

-be consistent with units – you use both W m$^{-2}$ and W/m$^2$

-line 53: delete the dot (.)

-line 98: Polar Regions => polar regions

-lines 99-100:
Why Singapore more digits than the mid-latitude site?

-line 104 – What do you mean by:
"…polar clouds that should be net cooling elements"

-line 140: 39m => 39 m

-line 273: …value for **theses** parametrizations (?)

-line 285-287:

-Why are the COD ranges different in Greenbelt and Singapore? Can that lead to differences in CRE?

-Why are the net CRE so different when comparing 20 sr with 30 sr results? Could you briefly mention that in the text?

---

## Referee Report (RR2)

**Review of Lolli et al., ACP -2016-980**

I thank the authors for including the suggested changes. I believe the manuscript can be published now.
Yet, I do still have 1 scientific comment (n6) and some comments on the writing style.

**Comments**

1.) *"Nighttime results will instead be considered as context to understanding net diurnal differences between the models when examining the GSFC dataset. "*

This isn't very clear, please rephrase.

2.)
I suggest restructuring section 3.2.

In the present form, the simultaneous description of 20 and 30 sr results does not help in the clarity of the message of the manuscript.

You first describe in lines 185-186 the 20 sr results, only later 30 sr results. But all of your further conclusions are based on 30 sr results, so I believe one should start with 30 sr, describe the results, and only later show the 20 sr results in a separate paragraph as some kind of a sensitivity test.

Moreover:
You kind of justify the choice of 30 sr for consistency with Sassen and Cho (1992), but not clearly stated why is valuable to look also at a different lidar ratio. I would just rephrase the sentence: *"The results here mirror the work…"*
It might not be clear to a reader what you meant with mirror? Please use more straightforward expressions for the convenience of the reader.

3.) lines 222-224
Please rephrase the following sentence:

*"A relative differencing here is impractical. Suffice however, this is a significant difference, and the sign of the net daytime forcing term is in direct question between the two."*

4.) line 226
I suggest the removing the first couple of words in the long and complicated sentence:
 *Relative to prior estimates of CP uncertainty compared with more complex models, a diurnal average would be likely to produce a different, and plausibly closer, relative agreement consistent with prior studies.*

5.) lines 296-297:
*Net daytime TOA CRE was evaluated versus cloud optical depth (COD) for steps of*

*0.03 (COD range: 0-1) at 20 sr and for steps of 0.1 at 30 sr (COD range: 0-3) for both the Singapore and Greenbelt, Maryland datasets.*

Please, be careful – you only show 20 sr for Singapore!

6.) Figure 7
I am surprised to see such a strongly negative net CRE. Could you briefly, maybe in one sentence, comment on that in the text at the end of the section 3.4?

---

## Author Response (AR2)

ACP-2016-980 – Second Interaction

Answers to the reviewer

The authors gratefully acknowledge the further suggestions of peer reviewer#2 assigned to this manuscript. We made all the proposed changes, improving the overall manuscript clarity.

Reviewer Comments in red
Author Responses in black

1.)Lines 17-18: "Specifically, CP simplifies...LW..."
You don't mention the simplification of the SW CRE calculation here, despite referring to the parameter y* (related to simplifications of SW flux calculations in CP model) further in the text?

The reviewer is completely right. We simply forgot to mention it into the abstract. Now it is expressly specified that CP simplifications apply both to SW and LW.

2.) You might want to mention the Greenbelt site in the abstract.

Done

3.) Chapter 3.1 – model sensitivities for -> daytime only? Please specify!
It would be also valuable to include information on LW and SW CRE component, not only NET.
I think that should be done already in this study (considering its simplicity), and not simply left as an outlook (which you state in lines 310-313).

We agree that the manuscript will be more complete adding SW and LW components to the study. Now, for each changed parameter in the sensitivity study, there is also how much is changed LW and SW CRE. Values have been added in Table 1.

4.) Chapters 3.2 and 3.3
I don't see what is the advantage of having both years separately, and not clustering the results together. If you keep them separately, you need to understand what caused a different value of the CP model bias (suggestion: Does it correlate with the surface temperature?)
And – if you keep two separate panels - Were the 2 years, meteorologically significantly different? Different ENSO phase, etc.?

Now we are persuaded that there is not variability for 2010 and 2011, so there is no reason to treat them separately. Now the analysis is performed on cirrus clouds of the whole period (2010-2011). We checked the mean, median and corresponding std without founding any difference in temperature patterns among the two years.

5.) Lines 205 – 214 :
I would suggest first stating the general conclusion (CP larger values for 40-60%) , followed by an example (1 W m-2 vs. 1.4-1.6 W m-2) .
Done

"..shows higher variability depending on the year…"
Yes, but year isn't the root cause of the change. What was different between 2010 and 2011? Maybe surface temperature?

Surface temperature is on average and std practically the same. For this reason we grouped both years together

6.) Lines 235-236:
I don't see any significant differences in the error between the 2 sites.

The sentence has been modified accordingly.

7.) Mention why CP2009 used σ* and k* and not simply k and σ.

Text has been modified accordingly. Now there is a description about the CP used parameters

8.) Lines 247-250:
Making a step further and trying to understand better what exactly is causing the error is something one could quickly look at. For instance, based on the Fig. 1 in CP2009 and your text lines 261-265, one would expect the error to be related to temperature, at least in the tropics. Your current conclusions of Chapter 3.4 do not add much to what already stated in CP2009.Therefore I would suggest (at least) one new sensitivity test in which you would if that leads to a better agreement with FLG model. (or select measurements atwarmest/coldest surface temperature conditions for both Singapore and Greenfield). This sensitivity test would be useful in better determining the conditions atwhich the CP model's bias is still at acceptably low levels (maybe those stated inCP2009 paper?).

That's a great suggestion that helps the manuscript to nail the main point. We perform net CRE calculation for Greenbelt for land surface temperature lower or equal to 288K and we found out that the two models are in a great agreement. We added a plot of this analysis. Definitely problems arise when temperatures are higher and more humidity is then in the atmosphere. We used 288K as lower limit and for this reason we picked up Greenbelt because in Singapore not so many points are available.

9.)
Figures 1-6:
I still don't see why not making a technical effort with Matlab to cluster the years 2010 and 2011. Couldn't you just make a sum of the two histograms/bar plots, as the x-axis values do not change?
This would sharpen the main points of the paper.

Now, for Singapore, years 2010 and 2011 are grouped together. Also plots are reduced, as for Singapore analysis there is only analysis over ocean and over land for the whole dataset

Minor comments

-be consistent with units – you use both W m-2 and W/m
changed accordingly

-line 53: delete the dot (.)
changed as suggested

-line 98: Polar Regions => polar regions
changed as suggested

-lines 99-100:
Why Singapore more digits than the mid-latitude site?
changed accordingly

-line 104 – What do you mean by:
"…polar clouds that should be net cooling elements"

-line 140: 39m => 39 m
changed as suggested

-line 273: …value for theses parameterizations (?)
now read:" value for these parameterizations

-line 285-287:

-Why are the COD ranges different in Greenbelt and Singapore? Can that lead to differences in CRE?
Greenbelt is using only 30sr, that has a COD going from 0 to 3, with 0.03 increment step. Also Singapore has the same for 30sr, while for 20sr the incremental step is 0.01 and COD is going from 0 to 1

-Why are the net CRE so different when comparing 20 sr with 30 sr results?
Could you briefly mention that in the text?

Supplementary rows have been added to explain why CRE is different at two LR

[revised manuscript text omitted]

---

## Author Response (AR3)

ACP-2016-980 – Third Interaction

Answers to the reviewer

Again, authors gratefully acknowledge the further suggestions of peer reviewer#2 assigned to this manuscript. We made all the proposed changes.

Reviewer Comments in red
Author Responses in black

1.) "Nighttime results will instead be considered as context to understanding net diurnal differences between the models when examining the GSFC dataset. " This isn't very clear, please rephrase.

The text has been rephrased as follows:" Those nighttime results presented within the analysis below will instead be considered as context for understanding net diurnally-averaged differences between the models specifically for the GSFC dataset."

2.)
I suggest restructuring section 3.2.
In the present form, the simultaneous description of 20 and 30 sr results does not help in the clarity of the message of the manuscript.You first describe in lines 185-186 the 20 sr results, only later 30 sr results. But all of your further conclusions are based on 30 sr results, so I believe one should start with 30 sr, describe the results, and only later show the 20 sr results in a separate paragraph as some kind of a sensitivity test.
Moreover:
You kind of justify the choice of 30 sr for consistency with Sassen and Cho (1992), but not clearly stated why is valuable to look also at a different lidar ratio. I would just rephrase the sentence: "The results here mirror the work…" It might not be clear to a reader what you meant with mirror? Please use more straightforward expressions for the convenience of the reader.

The manuscript has been changed as suggested by the reviewer and also the sentence has been rephrased

3.) lines 222-224 Please rephrase the following sentence: "A relative differencing here is impractical. Suffice however, this is a significant difference, and the sign of the net daytime forcing term is in direct question
between the two."

Text has been rephrased as: "A relative differencing here is impractical.  Suffice however, this is a significant difference, and the sign of the net daytime forcing term is uncertain when comparing the two"

4) line 226
I suggest the removing the first couple of words in the long and complicated sentence:
The thought here is that, Relative to prior estimates of CP uncertainty compared with
more complex models, a diurnal average would be likely to produce a different, and
plausibly closer, relative agreement consistent with prior studies.

Changed accordingly

5.) lines 296-297:
Net daytime TOA CRE was evaluated versus cloud optical depth (COD) for steps of
0.03 (COD range: 0-1) at 20 sr and for steps of 0.1 at 30 sr (COD range: 0-3) for both
the Singapore and Greenbelt, Maryland datasets. Please, be careful – you only show
20 sr for Singapore!

Changed accordingly

6.) Figure Figure 7
I am surprised to see such a strongly negative net CRE. Could you briefly, maybe
in one sentence, comment on that in the text at the end of the section 3.4?

A short paragraph has been added to explain the phenomenon.

[revised manuscript text omitted]